# Virtual Screening and ADMET Prediction to Uncover the Potency of Flavonoids from Genus *Erythrina* as Antibacterial Agent through Inhibition of Bacterial ATPase DNA Gyrase B

**DOI:** 10.3390/molecules28248010

**Published:** 2023-12-08

**Authors:** Abd. Wahid Rizaldi Akili, Ari Hardianto, Jalifah Latip, Afri Permana, Tati Herlina

**Affiliations:** 1Department of Chemistry, Faculty of Mathematics and Natural Science, Universitas Padjadjaran, Jatinangor 45363, Indonesia; abd21001@mail.unpad.ac.id (A.W.R.A.); a.hardianto@unpad.ac.id (A.H.); afri22001@mail.unpad.ac.id (A.P.); 2Department of Chemical Sciences, Faculty of Science and Technology, Universiti Kebangsaan Malaysia (UKM), Bangi 46300, Malaysia; jalifah@ukm.edu.my

**Keywords:** flavonoids, erythrina, structure-based virtual screening, ATPase DNA gyrase B, ADMET prediction

## Abstract

The emergence of antimicrobial resistance due to the widespread and inappropriate use of antibiotics has now become the global health challenge. Flavonoids have long been reported to be a potent antimicrobial agent against a wide range of pathogenic microorganisms in vitro. Therefore, new antibiotics development based on flavonoid structures could be a potential strategy to fight against antibiotic-resistant infections. This research aims to screen the potency of flavonoids of the genus Erythrina as an inhibitor of bacterial ATPase DNA gyrase B. From the 378 flavonoids being screened, 49 flavonoids show potential as an inhibitor of ATPase DNA gyrase B due to their lower binding affinity compared to the inhibitor and ATP. Further screening for their toxicity, we identified 6 flavonoids from these 49 flavonoids, which are predicted to have low toxicity. Among these flavonoids, erystagallin B (**334**) is predicted to have the best pharmacokinetic properties, and therefore, could be further developed as new antibacterial agent.

## 1. Introduction

Throughout the human history, the discovery and development of antibiotics is one of the greatest inventions to combat infectious diseases. Unfortunately, due to improper the use of antibiotics, have caused the emergence of drug-resistant microbes [1]. According to a 2019 report from the World Health Organization (WHO), antimicrobial resistance is responsible for 700,000 deaths worldwide, with the death count expected to rise to 20 million by 2050, and could cost up to US$ 2.9 trillion [2]. This shows that antimicrobial resistance is a serious problem and needs to be addressed immediately [3].

A target-based approach is one efficient and relatively inexpensive approach in drug discovery and development research [4]. Target-based approaches emerged in line with the development of chemistry and pharmacology in the early 20th century. In recent decades, drug discovery with target-based approaches has become a major paradigm in the early stages of drug discovery; this approach has increased the screening capacity of drug compounds and improved the rationalization of drug discovery [5].

Bacterial DNA gyrase is one protein that has been validated as a target protein in the development of antimicrobial drugs [6]. There are several binding sites on DNA gyrase that can be targeted for antimicrobial discovery and development; DNA gyrase subunit B is one of them. DNA gyrase subunit B contains an ATPase domain, which is the binding site for ATP molecules to be hydrolyzed to produce the energy needed for DNA gyrase to carry out its catalytic function [7]. The protein structure of the ATPase domain of DNA gyrase B has been widely studied so that this domain can be used to develop inhibitors with high binding affinity [8]. Moreover, if the ATPase domain undergoes mutations to reduce interactions with ATPase inhibitor drugs, then these mutations will also cause DNA gyrase to lose its function to hydrolyze ATP [9](, making ATPase DNA gyrase B a potential target for the development of antimicrobial agent, especially to fight against resistant microbes.

Natural products are an important source of new antimicrobial agents that can be utilized against microbes that have developed resistance [10]. Flavonoids are among these natural products that have shown to have potential as an ATPase DNA gyrase B inhibitor [11]. For instance, quercetin is reported to bind to the 24 kDa fragment of *E. coli* DNA gyrase B and inhibit DNA gyrase B ATPase activity. The binding side of quercetin is known to overlap with the ATP binding pocket and competes with both ATP and novobiocin [12]. One of the plants rich in flavonoid content is the *Erythrina* genus; so far, at least 409 flavonoids have been isolated from this genus [13,14]. Among these flavonoids, many have been reported for their potent antibacterial activity against wide range of bacterial. Nevertheless, their potential as ATPase DNA gyrase B inhibitor have not yet been reported, making the exploration of the potential of flavonoids, especially those obtained from genus Erythrina, as ATPase inhibitors of DNA gyrase B an interesting topic to be studied. Thus, this research aimed to study the potency of flavonoids from genus Erythrina as inhibitors of DNA gyrase B, along with their toxicity and pharmacokinetic prediction through the use of an in silico study.

## 2. Results

### 2.1. Structure-Based Virtual Screening

The structure of the 378 flavonoids of genus Erythrina was subjected to virtual screening against the E. coli DNA Gyrase B 24 kDa ATPase domain. These flavonoids include flavones (eighteen compounds); flavonols (three compounds); flavanones (ninety-one compounds); chalcones (fourteen compounds); isoflavans (eleven compounds); isoflav-3-enes (five compounds); isoflavanones (forty-three compounds); isoflavones (ninety-nine compounds); pterocarpans (forty-two compounds); 6α-hydroxypterocarpans (fourteen compounds); pterocarpenes (seven compounds); coumestans (five compounds); 3-arylcoumarins (three compounds); coumaronochromone (one compounds); 2-arylbenzofurans (fifteen compounds); 3-aryl-2,3-dihydrobenzofurans (two compounds); and bioflavonoid (one compound).

Based on the binding affinity, among the 378 flavonoids that were screened for their potency as ATPase DNA gyrase B inhibitor, 49 compounds were found to have a lower binding energy than the co-crystal ligand, which is an inhibitor of the E. coli DNA Gyrase B 24 kDa ATPase domain (Table 1). The binding energy of the co-crystal ligand was −8.853 kcal/mol, whereas the binding energy of these 49 flavonoids were in the range of −10.36 to −8.861 kcal/mol. In addition, the binding energy of these 49 flavonoids were also lower than the binding energy of ATP, which is a natural substrate of the ATPase domain of DNA Gyrase, making these flavonoids a potential inhibitor of DNA Gyrase B 24 kDa ATPase domain. The binding affinity of all 378 flavonoids is provided in the Appendix A.

Among the 49 compounds, which exhibit a lower binding energy than the co-crystal ligand, some of these compounds have been reported to exhibit significant antibacterial activity. Eryzerin C (**131**) is reported to exhibit very strong activity against *S. aureus*, *S. epidermidis*, *E. coli*, and *P. aeruginosa*, with an MIC of 5, 2, 5, and 5 μg/mL [15]. Eryzerin D (**130**) is reported to show strong antibacterial activity against MRSA, with an MIC 12 μg/mL [16]. Erypoegin H (**347**) shows strong antibacterial activity against MRSA, with an MIC of 12.5 μg/mL [17]. Erysubin F (**193**) shows good antibacterial activity against MRSA, with an MIC of 100 μg/mL [18]. Erystagallin A (**332**) exhibits very strong antibacterial activity against S. aureus, MRSA, and VRSA, with an MIC of 1.56, 0.79, and 1.56, respectively [19].

### 2.2. Toxicity Screening

The toxicity aspect of a drug candidate compound is important to consider in the early stages of drug development [20]. Due to increasing number of large toxicology databases available, computational toxicity prediction has become very useful for weeding out compounds that have a high likelihood of failing clinical trials, especially in the early stages of the drug development process. There are numerous toxic end points that have been predicted in silico. The most commonly investigated include hepatotoxicity, cardiotoxicity, and mutagenicity data end points [21]. Therefore, after identifying flavonoid structures, which have potential as an inhibitor of the DNA Gyrase B 24 kDa ATPase domain, the next step is to screen these flavonoids for their potency of being toxic to human health.

The toxicity potency of the 49 flavonoids, which are potentially inhibitors of ATPase DNA gyrase B, were screened through an in silico study using ADMETlab 2.0 webserver [22]. These flavonoids were screened for their potency as hERG blockers, human hepatotoxic, DILI, mutagenic, carcinogenicity, and respiratory toxicity. The results are summarized in Table 2.

### 2.3. Pharmacokinetics

Pharmacokinetics can be defined as the study of the dynamic movement of foreign chemicals (xenobiotics) in the body, including their absorption, distribution, transformation/metabolism, and excretion (ADME) parameters [23]. The study of the pharmacokinetic profile of a drug candidate is an important stage in drug development; the prediction of the pharmacokinetic profile in the early stages of drug design and development can help avoid failure in the future clinical trials of drug candidates [24]. In addition to toxicity issue, poor pharmacokinetics properties are believed to be the major reasons for discontinuing the development process of drug candidates [25]. Therefore, for flavonoids with a low to moderate toxicity end point, we continue to evaluate their pharmacokinetic parameters in silico (Table 3).

## 3. Discussion

The emergence of antimicrobial resistance due to the widespread and inappropriate use of antibiotics has now become a global health challenge [26], making it clear that the research and development of new antimicrobial agent is needed [27]. Natural products continue to play fundamental role in the process of drug discovery and development [28]. They have long been the primary source of novel drugs and method to fight against infectious diseases. In fact, almost 75% of today’s antibiotics are derived from natural products [29]. Natural-derived secondary metabolites, particularly flavonoids, have long been reported to have strong in vitro antibacterial effects against a variety of pathogenic microbes. Therefore, the development of new antibiotics based on flavonoid structures could be a potential strategy to fight against antibiotic-resistant infections [30].

The genus *Erythrina* is one of many higher plantthat is rich in flavonoids. About 378 flavonoids have been isolated from this genus. Among these flavonoids, some have been reported for their antibacterial activity [12]. Numerous studies have found that flavonoids exert their antibacterial activities through a variety of different mechanism of action [31], one of which is through the inhibition of ATPase DNA gyrase B [30]. The 24 kDa ATPase domain of DNA gyrase B has been reported to be a potential target protein for antibiotic development [32]. Therefore, in this research, we try to screen flavonoids from genus Erythrina for their potency as inhibitors of ATPase DNA gyrase B through an in silico study. In our research, a total of 378 flavonoids isolated from the genus Erythrina was subjected to virtual screening against the ATPase domain of DNA gyrase B by employing molecular docking. Among these 378 flavonoids, 49 flavonoids exhibit a lower binding affinity than the co-crystal ligand and ATP molecule, indicating their potential as potent inhibitors of the ATPase domain of DNA gyrase B. For further screening, these 49 flavonoids were then subjected to toxicity screening, from which it was found that six of these flavonoids were the least toxic, with a low to medium risk of being toxic. Subsequently, we conducted a pharmacokinetic property evaluation on these six flavonoids and our analysis revealed that erystagallin B (**334**) is predicted to exhibit the most favorable pharmacokinetic properties (Figure 1).

### 3.1. Structure-Based Virtual Screening

Prior to structure-based virtual screening, the molecular docking protocol validation was conducted. This is an important step for an in silico docking experiment because it can increase the accuracy and reliability of the experiment [33]. A molecular docking protocol is valid if the RMSD of the docked ligand is less than 2 Å [34]. In our experiment, we obtained an RMSD value of 1.72 Å; therefore, our docking protocol is valid and can be employed to virtually screen the potency of flavonoids from genus Erythrina as an inhibitor of bacterial ATPase DNA gyrase B.

The binding affinity as well as the binding pose of a ligand and enzyme generated from molecular docking study are crucial pieces of information for computer-aided drug design. In the AutoDock Vina docking score, the more negative the score is, the stronger the binding affinity of the ligand and the enzyme [35]. Therefore, from the result of molecular docking, judging from their binding affinity, among the 378 flavonoids that were screened, 49 flavonoids showed a lower binding affinity compared to the co-crystal ligand and ATP molecule (Table 1). This suggests that these compounds could be potential inhibitors of ATPase DNA gyrase B. The co-crystallized ligand showed hydrogen bond interaction with Asp73. This interaction is an important interaction, as it observed in many known ATP competitive inhibitors of DNA gyrase B [8]. The same interaction with Asp-73 is also found with ATP molecules, but it is not seen with compound 287 (i.e., compound that exhibit lowest binding affinity), suggesting different binding mode of compound 287 and the co-crystallized ligand (Figure 2).

### 3.2. Toxicity Prediction

For further screening, these 49 flavonoids were subjected to an in silico toxicity screening to predict their toxicity properties, including hERG blocker, human hepatotoxicity, drug-induced liver injury (DILI), mutagenicity, carcinogenicity, and respiratory toxicity. Among these 49 flavonoids, only compound 287 is predicted to have a high risk of being an hERG blocker. A total of 29 compounds are predicted to have a high risk of being hepatotoxic, 20 compounds are predicted to have high risk of causing drug-induced liver injury (DILI), 2 compounds are predicted to have a high risk of being mutagenic, 15 compounds are predicted to have a high risk of being mutagenic, and 21 compounds are predicted to have a high risk of causing respiratory toxicity. Only 6 compounds of 49 flavonoids being screened are predicted to have a low to medium potency of being toxicfor the tested toxicity parameter (Table 2). These compounds are lonchocarpol C (**74**), erystagallin B (**334**), lonchocarpol D (**75**), ficuisoflavone (**223**), 4′-Hydroxyisoflavone-7-O-β-d-glucopyranoside (**286**), and erythraddison IV (**174**) (Figure 3).

### 3.3. Pharmacokinetics Prediction

The pharmacokinetics and toxicity properties of a potential drug candidate should be taken into account as early as possible in order to lower failure rates in the clinical phase of drug discovery since undesirable pharmacokinetics and toxicity are key factors in the costly late stage of drug development failure [36]. Therefore, after identifying the potential toxicity of the screened flavonoids, we continued to evaluate the pharmacokinetics properties of the six flavonoids with low to medium potency and toxicity through an in silico study.

#### 3.3.1. Absorption

Drug absorption describes how well a unmetabolized drug is transported from the site of drug administration into the bloodstream [37]. The drug absorption pharmacokinetic parameter for the six flavonoids include Caco-2 permeability, MDCK permeability, Pgp inhibitor and substrate, and human intestinal absorption. The Caco-2 and MDCK in vitro permeability assays are commonly employed to provide permeability measurements and to investigate the contribution of active transporters that may influence the distribution of drug candidates and other xenobiotics. [38]. The Caco-2 cell monolayer permeability model has long been used to evaluate a drug’s permeability through the human intestinal epithelial cell barrier, thereby providing information about the potential bioavailability of a drug in the human body [39].

In ADMETlab model prediction, the predicted Caco-2 permeability of a given compound is considered proper if it has predicted value >−5.15 log cm/s. For MCDK permeability, a compound with Papp >20 × 10^−6^ cm/s is considered to have a high passive MDCK, whereas for medium permeability, the cutoff is 2–20 × 10^−6^ cm/s, and for low permeability, it is <2 × 10^−6^ cm/s [22]. Referring to Table 2, compounds **74**, **334**, **75**, **223**, and **174** are predicted to have proper Caco-2 permeability, whereas compound **286** is predicted to have poor Caco-2 permeability. Nevertheless, each of these six compounds are predicted to have medium MDCK permeability.

Another parameter that plays major role in drug absorption is P-glycoprotein transporter [40]. P-glycoprotein (P-gp), commonly known as multidrug-resistance protein 1 (MDR1), is a protein that transports substrates into the bile, urine, and gastrointestinal system. P-gp inhibitors are hypothesized to improve medication absorption, whereas P-gp substrates diminish drug absorption by being ejected from cells when used together. As a result, understanding whether a chemical is a P-gp substrate or an inhibitor is crucial in the early phases of drug development [41]. According to ADMETlab model prediction, in terms of being a Pgp-inhibitor, compounds **74**, **334**, and **75** are predicted to have a medium probability of being a P-gp inhibitor, whereas compounds **223**, **286**, and **174** are predicted to have a high probability of being P-gp-inhibitor. On the other hand, in terms of being a Pgp-substrate, compound **74**, **75**, **286**, and **174** are predicted to have a low probability of being a Pgp-substrate, whereas compounds **334** and **223** are predicted to have a high probability of being a Pgp-substrate. In terms of human intestinal absorption, compounds **74**, **334**, **75**, **223**, and **174** are predicted to have high intestinal absorption, whereas compound **286** has medium intestinal absorption.

Skin permeability is the pharmacokinetic parameter that determines the rate and extent of drug absorption through the skin. Therefore, understanding skin permeability is crucial for the development of effective topical and transdermal drug delivery systems [42]. The skin permeability prediction is expressed as logKp (cm/h). A compound is predicted to have low skin permeability if the compound in question has a logKp value of >−2.5 [43]. Therefore, compounds **74**, **334**, **75**, **223**, **286**, and **174** are predicted to have high skin permeability.

#### 3.3.2. Distribution

Drug distribution is important parameter to describe how an unmetabolized drug is dispersed from the blood stream to various tissues of the body [44]. The drug distribution parameter being evaluated in silico include plasma protein binding, volume distribution, BBB penetration, and fraction unbound. The evaluation of the distribution property of a potential drug is crucial because it plays important role in the exposure of the target organ to the drug [45].

In the blood stream, drugs exist in an equilibrium state between a bound and unbound form with plasma protein. The degree to which the drug reversibly attaches to plasma protein is referred as plasma binding protein (PBP). The fraction of drug that binds to plasma protein is an essential pharmacokinetic property that is closely related to the drug’s absorption, distribution, metabolism, excretion, and toxicity. Since only the unbound portions of the drug can be transported into tissues, the drug candidate with an inappropriate PPB value could be discarded in the final stage of drug discovery [46]. According to ADMETlab model prediction, a compound is considered to have a proper PPB if it has predicted value <90%, and therefore, among the six flavonoids, only compound **334** is predicted to have a proper PPB value.

The volume of distribution (Vd) represents the drug’s propensity to either remain in the plasma or to distribute into different tissue compartments. A high Vd value indicates that a drug is more likely to migrate out of the plasma and enter into the body’s extravascular compartments, therefore requiring a greater dosage in order to achieve the desired plasma concentration. Conversely, a low Vd value indicates that a drug is more likely to remain in the plasma, therefore it requires lower dosage in order to achieve the desired plasma concentration [47]. According to ADMETlab model prediction, a compound with a predicted VD value in the range of 0.04–20 L/kg is considered to have a proper VD, and therefore, each of the six flavonoids are predicted to have proper volume of distribution value.

The blood–brain barrier (BBB) is a diffusion barrier that plays a crucial role in protecting brain function by preventing most compounds, including drugs, from the blood to pass into the brain [44]. In order to maintain central neural system (CNS) homeostasis, the BBB blocks 98% of exogenous chemicals to enter the CNS. Therefore, determining a compound’s BBB permeability is a prerequisite for screening chemicals/bio-molecules that may have effects on the CNS. According to ADMETlab model prediction, each of the six flavonoids are predicted to have low penetration on the blood–brain barrier (BBB).

The binding of drugs to plasma proteins limits their free and active concentrations, thereby altering their pharmacokinetic properties [48]. In pharmacokinetic and pharmacodynamic studies, the fraction unbound in plasma (Fu) is a significant determinant of drug efficacy. This is because only the unbound (free) drugs are capable of diffusing between plasma and tissues and therefore capable of interacting with pharmacological target proteins, including receptors, channels, and enzymes. Therefore, it is necessary to accurately predict fraction unbound in plasma, especially in the low-value ranges, during the course of drug development [49]. According to ADMETlab model prediction, a compound is considered to have a high Fu if it has a predicted value >20%, medium Fu if in the range of 5–20%, and low if <5%. Therefore, compounds **334** and **286** are predicted to have a medium Fu, whereas compound **74**, **75**, **223**, and **174** are predicted to have a low Fu.

#### 3.3.3. Metabolism

Most drugs are xenobiotics, i.e., foreign chemicals that are not naturally synthesized by the body. In the body, xenobiotics are modified through numerous mechanisms in order to reduce their toxicity and to allow them to be readily available for excretion. The modification of xenobiotics into their metabolites in the body is known as metabolism or metabolic biotransformation [50]. The drug metabolism process in the body can be divided into two broad categories based on the chemical nature of biotransformation, i.e., phase I (oxidative reaction) and phase II (conjugative reaction). The human cytochrome P450 family (phase I enzymes) consist of 57 isozymes, which metabolize nearly two-thirds of known pharmaceuticals in humans, with 80% of this being attributable to five isozymes, namely 1A2, 3A4, 2C9, 2C19, and 2D6 [23]. The inhibition and induction of cytochrome P450 (CYP) enzymes are central mechanisms that result in clinically significant drug–drug interactions (DDI) [51]. According to ADMETlab model prediction, compound **74** is predicted to be an inhibitor of CYP2C19, CYP2C9, and CYP2D6, and this compound is predicted to be a substrate of CYP2C9. Compound **334** is predicted to be both an inhibitor and substrate of CYP2C19 and CYP2C9 as well as an inhibitor of CYP2D6. Compound **75** is predicted to be both an inhibitor and substrate of CYP2C9, as well as an inhibitor of CYP2C19, and CYP2D6. Compound **223** is predicted to be both an inhibitor and substrate of CYP2C9 and CYP2D6, as well as an inhibitor of CYP1A2 and CYP2C19. Compound **286** is predicted to be both an inhibitor and substrate of CYP2D6. Compound **174** is predicted to be both an inhibitor and substrate of CYP1A2, CYP2C9, and CYP2D6, as well as an inhibitor of CYP2C19 and CYP3A4.

#### 3.3.4. Excretion

Drug excretion consists of a variety of mechanisms that remove a given drug and/or its metabolites from the body and is the final step in the ADME (absorption, distribution, metabolism, and excretion) process. The excreted drugs are either removed in their unmetabolized form or eliminated after metabolic biotransformation [52]. The drug excretion parameter being evaluated in silico include drug clearance and half-life (t_1/2_).

Drug clearance is defined as the amount of drug removed from the plasma in the vascular compartment per unit of time. Total clearance indicates drug removal from the central compartment without regard for the mechanism behind this process. According to ADMETlab model prediction, a compound is considered to have high clearance if the compound has a predicted value >15 mL/min/kg, moderate clearance if predicted value 5–15 mL/min/kg, and low clearance if it is <5 mL/min/kg [23]. Therefore, compounds **334** and **75** are predicted to have high clearance; compounds **74**, **223**, and **174** are predicted to have moderate clearance; whereas compound **286** is predicted to have low clearance.

## 4. Materials and Methods

### 4.1. Flavonoids Structure Preparation

The structure of the 378 flavonoids were retrieved from a literature review reported by Fahmy et al., in 2018 [12]. The protonated state of these flavonoids were predicted using Chemaxon MarvinSketch followed by structure optimization using Merck molecular force field (MMFF94). The generated 3D structures were then prepared for molecular docking, including adding H atoms, adding Gastaiger charges, and determining rotatable bonds that will be explored during the docking process.

### 4.2. Protein Preparation

The 3D X-ray crystal structure of the *E. coli* DNA Gyrase B 24 kDa ATPase domain (PDB ID 5MMN) was retrieved from Protein Data Bank (PDB) (https://www.rscb.org, accessed on 5 May 2023). The co-crystalized ligand was separated from the 3D structure of the protein and followed by the removal of explicit water molecules to make the protein a nascent receptor. Subsequently, hydrogen atoms and charges were added to the protein.

### 4.3. Molecular Docking Validation

Protein and co-crystalized ligand structures that have been prepared were re-docked using Autodock Vina 1.2.0 [53]. The gridbox parameter was set to cover the co-crystalized ligand and the largest flavonoid structure to be docked. The co-crystalized coordinate and gridbox parameters were copied to a configuration file. The re-docking procedure was run through the command prompt. After molecular docking was completed, all the docking poses were separated, and the pose with the lowest binding energy was opened in Biovia discovery and superimposed with the co-crystalized ligand file before molecular docking. Then, the RMSD value was calculated.

### 4.4. Virtual Screening

The prepared three-dimensional structure of the 378 flavonoids were docked in the *E. coli* DNA Gyrase B 24 kDa ATPase domain in the same way as molecular docking validation.

### 4.5. ADMET Prediction

ADMET prediction was conducted through the ADMETlab 2.0 web server (https://admetmesh.scbdd.com, accessed on 5 May 2023). A list of SMILES of the flavonoid structures with lower binding affinity compared to the co-crystal ligand and ATP was prepared. These SMILES were then copied to the ADMETlab 2.0 web server to predict their pharmacokinetics and toxicity properties.

## 5. Conclusions

In this research, we have conducted a virtual screening of 378 flavonoids from the genus Erythrina that are potential ATPase DNA gyrase B inhibitors. Among these 378 flavonoids, we identified 6 flavonoids which are have potential as inhibitors of ATPase DNA gyrase B and are predicted to have low to moderate toxicity, i.e., lonchocarpol C (**74**), erystagallin B (**334**), lonchocarpol D (**75**), ficuisoflavone (**223**), 4′-Hydroxyisoflavone-7-O-β-d-glucopyranoside (**286**) and erythraddison IV (**174**). Among these compounds, erystagallin B (**334**) is predicted to have the best pharmacokinetic properties among the other compounds that were evaluated. Therefore, this compound could be further developed as a new antibacterial agent.

## Figures and Tables

**Figure 1 molecules-28-08010-f001:**
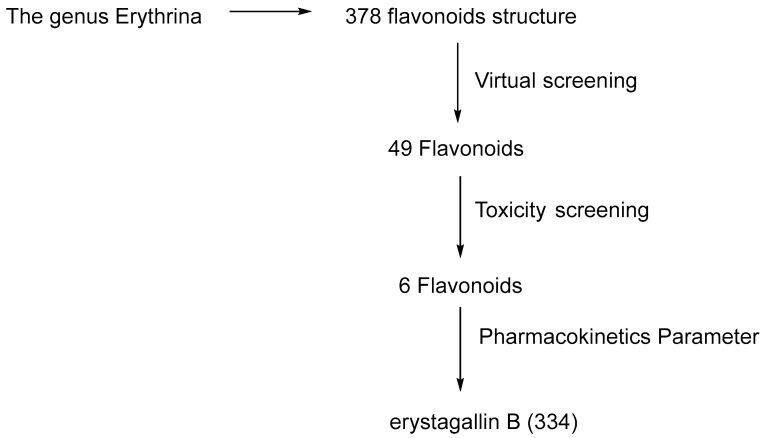
Diagram on virtual screening of flavonoids from genus Erythrina as possible ATPase DNA gyrase B inhibitor.

**Figure 2 molecules-28-08010-f002:**
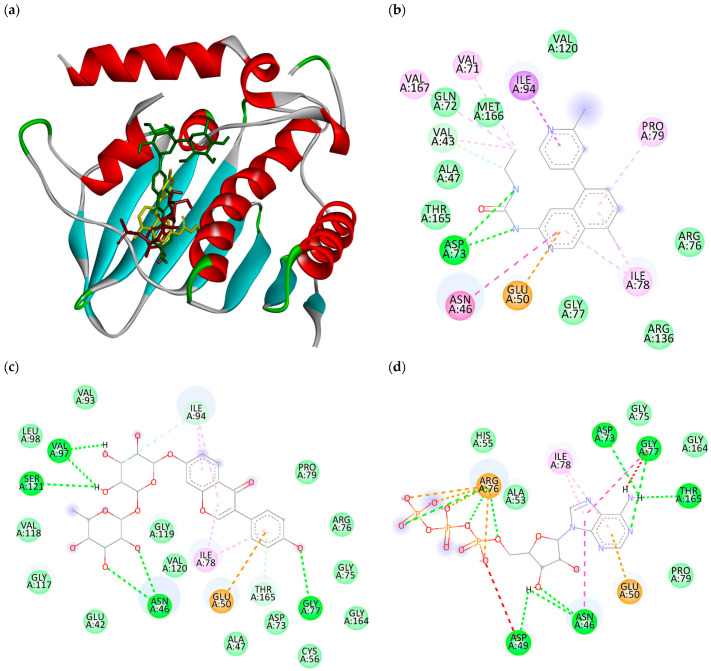
Superimposed image of co-crystallized ligand (yellow), compound 287 (green), and ATP (red) in the ATPase domain of DNA gyrase B (**a**), interaction of co-crystallized ligand (**b**), compound 287 (**c**), and ATP (**d**) with amino acids in the ATPase domain of DNA gyrase B.

**Figure 3 molecules-28-08010-f003:**
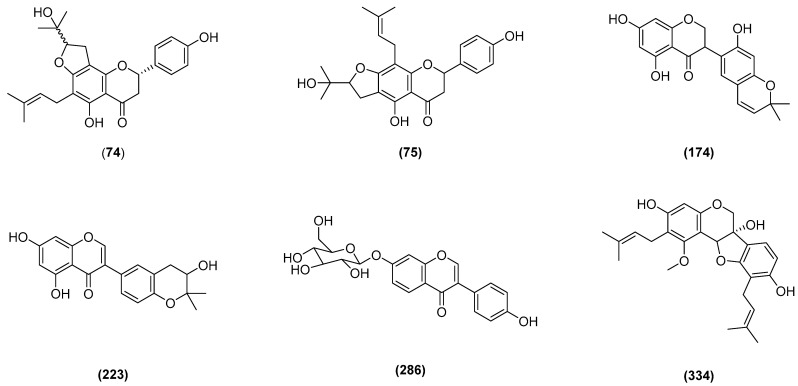
Structure of compounds **74**, **334**, **75**, **223**, **286**, and **174**.

**Table 1 molecules-28-08010-t001:** List of 49 flavonoids with low binding energy on ATPase DNA gyrase B.

Compound Name	Flavonoid Subclass	Binding Affinity (Kcal/mol)
4′-Hydroxyisoflavone-7-O-α-L-rhamnosyl/(1 → 6)-β-d-glucopyranoside (**287**)	Isoflavone	−10.36
Erypoegin J (**339**)	6α-Hydroxypterocarpan	−10.15
Sigmoidin K (**354**)	Coumestans	−9.941
Erysubin E (**341**)	6α-Hydroxypterocarpan	−9.825
Demethylerystagallin A (**333**)	6α-Hydroxypterocarpan	−9.79
Sigmoidin K (**345**)	Coumestan	−9.771
Apigenin-7-O-rhamnosyl-6-C-glucoside (**14**)	Flavone	−9.615
Kaempferol-3-O-(2ʹʹ-O-β-d-glucopyranosyl-6ʹʹ-O-α-L-rhamnopyranosyl-β-d-glucopyranoside) (**20**)	Flavone	−9.599
Lonchocarpol C (**74**)	Flavanone	−9.489
Erythribyssin L (**325**)	Pterocarpans	−9.469
Eryzerin C (**131**)	Flavanone	−9.411
Erythribyssin O (**351**)	Pterocarpene	−9.377
2(S)-5,7-Dihydroxy-[(5′′,6′′:3′,4′)- (2′′,2′′-dimethylpyrano)- (5ʹʹʹ,6′′′:5′,6′)]-(2′′′,2′′′- dimethylpyrano)flavanone (**70**)	Flavanone	−9.373
Sigmoidin E (**64**)	Flavanone	−9.352
(2S)-5,7-Dihydroxy-5′-prenyl-2ʹʹ-(4ʹʹ-hydroxyisopropyl)- dihydrofurano [1′′,3′′:3′,4′] flavanone (**94**)	Flavanone	−9.287
Isosojagol (**355**)		−9.275
Isolupabigenin (**210**)	Isoflavone	−9.244
2(S)-5,7-Dihydroxy-5′-prenyl- [2′′,2′′-(3′′-hydroxy)- dimethylpyrano]-(5′′,6′′:3′,4′) flavanone (**88**)	Flavanone	−9.228
Neocyclomorusin (**7**)	Flavone	−9.208
Fuscaflavanones B (**79**)	Flavanone	−9.184
(2S)-5,7,5′-Trihydroxy-2ʹ′-(4′′- hydroxyisopropyl)-dihydrofurano [1′′,3′′:3′,4′]flavanone (**96**)	Flavanone	−9.149
Iespedezaflavanone B (Euchrestaflavanone A) (**52**)	Flavanone	−9.124
Eryzerin D (**130**)	Isoflavans	−9.124
Erystagallin B (**334**)	6α-Hydroxypterocarpans	−9.126
5,3ʹ-Dihydroxy-4ʹ-methoxy-5ʹ-γ,γ-dimethylallyl-2ʹʹ,2ʹʹ-dimethylpyrano [5,6:6,7] isoflavanone (**178**)	Isoflavanone	−9.096
(2S)-5,7,5′-Trihydroxy-2′′-(4′′- hydroxyisopropyl)-3′′-hydroxy-dihydrofurano [1′′,3′′:3′,4′] flavanone (**98**)	Flavanone	−9.094
Lonchocarpol D (**75**)	Flavanone	−9.068
Vogelin G (**222**)	Isoflavone	−9.061
Erysenegalensein F (**264**)	Isoflavones	−9.061
Erypoegin H (**347**)	Pterocarpene	−9.058
Ficuisoflavone (**223**)	Isoflavone	−9.048
Fuscaflavanones A2 (**78**)	Flavanone	−9.043
4′-Hydroxyisoflavone-7-O-β-d-glucopyranoside (**286**)	Isoflavone	−9.021
Corylin (**201**)	Isoflavone	−8.971
Erythribyssin G (**40**)	Flavanone	−8.966
Erylivingstone K (**137**)	Isoflavan	−8.965
Erysubin F (**193**)	Isoflavone	−8.951
2ʹ,7-Dihydroxy-3ʹ -(3-methylbut-2- enyl)-2ʹʹʹ,2ʹʹʹ-dimethylpyrano [5ʹʹ,6ʹʹ:4ʹ,5ʹ]isoflavan (**138**)	Isoflavan	−8.946
Lonchocarpol A (Senegalensein) (**73**)	Flavanone	−8.941
(2S)-5,7,5′-Trihydroxy-6′-prenyl-2′′-(4′′-hydroxyisopropyl)-3′′- hydroxy-dihydrofurano [1′′,3′′:4′,5′] flavanone (**99**)	Flavanone	−8.941
Erystagallin A (**332**)	6α-Hydroxypterocarpans	−8.937
2(S)-5,7-Dihydroxy- [2′′,2′′-(3′′,4′′-dihydroxy)- dimethylpyrano]-(5′′,6′′:3′,4′) flavanone (**90**)	Flavanone	−8.933
Erythraddison IV (**174**)	Isoflavanone	−8.922
Glabrol (**24**)	Flavanone	−8.914
Vogelin I (**278**)	Isoflavone	−8.888
Fuscaflavanones A1 (**77**)	Flavanone	−8.887
Erysenegalensein L (**267**)	Isoflavone	−8.865
2(S)-5,7-Dihydroxy-5′-prenyl [2′′,2′′-(3′′,4′′-dihydroxy)- dimethylpyrano]-(5′′,6′′:3′,4′) flavanone (**91**)	Flavanone	−8.863
Erylatissin B (**200**)	Isoflavone	−8.861
1-ethyl-3-[8-methyl-5-(2-methyl-pyridin-4-yl)-isoquinolin-3-yl]-urea	Co-crystal ligand	−8.853
ATP	Natrual substrate	−7.498

**Table 2 molecules-28-08010-t002:** Toxicity prediction of flavonoids with lower binding energy than the co-crystal ligand.

Compounds	Toxicity Parameter					
	hERG Blockers	H-HT	DILI	Mutagenic	Carcinogenicity	Respiratory Toxicity
**287**	A	A	B	B	C	A
**339**	C	C	A	B	C	C
**354**	A	C	C	A	B	A
**341**	B	C	A	B	C	C
**333**	A	C	A	B	A	B
**345**	A	C	C	B	A	C
**14**	A	B	C	A	A	A
**20**	B	A	C	B	A	A
**74**	A	B	B	A	A	B
**131**	A	B	A	A	C	C
**70**	A	C	C	A	C	C
**64**	A	C	B	A	B	C
**94**	A	C	B	A	A	C
**355**	A	C	C	A	B	A
**210**	A	C	C	A	A	A
**88**	A	C	B	A	B	C
**7**	A	C	C	B	C	B
**79**	A	C	C	A	B	B
**96**	A	A	C	A	B	C
**52**	A	C	C	A	A	C
**130**	B	C	A	A	B	C
**334**	A	B	A	A	A	B
**178**	A	C	B	A	B	A
**98**	A	A	C	A	A	B
**75**	A	A	B	A	A	A
**222**	A	C	A	A	A	A
**264**	A	B	C	A	C	B
**347**	A	C	C	C	A	B
**223**	A	A	A	A	B	A
**78**	A	C	C	A	C	C
**286**	A	A	B	A	B	A
**201**	B	B	B	A	C	C
**40**	A	C	B	A	B	C
**137**	A	B	A	A	C	C
**193**	A	C	B	A	A	C
**138**	B	C	A	A	B	A
**73**	A	C	C	A	A	C
**99**	A	C	C	A	A	C
**332**	B	C	A	B	A	B
**90**	A	B	A	A	C	C
**174**	A	A	B	A	B	B
**24**	A	C	B	A	A	C
**278**	A	B	B	A	C	A
**77**	A	C	C	C	C	C
**267**	A	C	C	A	C	A
**91**	A	C	C	A	B	A
**200**	A	B	A	A	C	B

A: Low risk; B: moderate risk; C: high risk; H-HT: human hepatotoxic; DILI: drug-induced liver injury.

**Table 3 molecules-28-08010-t003:** Pharmacokinetics prediction of compounds **74**, **334**, **75**, **223**, **286**, and **174**.

Pharmakokinetics Parameter	Compounds					
	74	334	75	223	286	174
**Adsorption**						
Caco−2 permeability	−4.832	−4.887	−4.839	−4.848	−6.104	−4.947
MDCK permeability	1.4 × 10^−5^	1.3 × 10^−5^	1.3 × 10^−5^	1 × 10^−5^	1.2 × 10^−5^	1.7 × 10^−5^
Pgp−inhibitor	Medium probability	Medium probability	Medium probability	High probability	High probability	High probability
Pgp−substrate	Low probability	High probability	Low probability	High probability	Low probability	Low probability
Human intestinal absorption	High	High	High	High	Medium	High
Skin permeation	−2.859	−2.737	−2.938	−2.738	−2.735	−2.762
**Distribution**						
Plasma protein binding (PBP)	97.205%	87.618%	98.748%	98.416%	90.990%	99.914%
Volume distribution (VD)	0.926	2.476	0.654	0.507	0.979	0.440
BBB penetration	Low penetration	Low penetration	Low penetration	Low penetration	Low penetration	Low penetration
Fraction unbound	4.677%	13.018%	2.757%	1.610%	6.015	0.891%
**Metabolism**						
CYP1A2 inhibitor	No	No	No	Yes	No	Yes
CYP1A2 substrate	No	No	No	No	No	Yes
CYP2C19 inhibitor	Yes	Yes	Yes	Yes	No	Yes
CYP2C19 substrate	No	Yes	No	No	No	No
CYP2C9 inhibitor	Yes	Yes	Yes	Yes	No	Yes
CYP2C9 substrate	Yes	Yes	Yes	Yes	No	Yes
CYP2D6 inhibitor	Yes	Yes	Yes	Yes	Yes	Yes
CYP2D6 substrate	No	No	No	Yes	Yes	Yes
CYP3A4 inhibitor	No	No	No	No	No	Yes
CYP2A4 substrate	No	No	No	No	No	No
**Excretion**						
Clearance	12.099	15.185	15.185	6.230	2.363	7.857

## Data Availability

No new data were created or analyzed in this study. Data sharing is not applicable to this article.

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
