# Peer review of "Virtual Screening and ADMET Prediction to Uncover the Potency of Flavonoids from Genus Erythrina as Antibacterial Agent through Inhibition of Bacterial ATPase DNA Gyrase B"

_molecules, 2023, doi:10.3390/molecules28248010_

Round 1
Reviewer 1 Report
The topic of this manuscript is interesting and important. The overall presentation is acceptable but, in my opinion, some improvements could be made. I would recommend a more detailed explanation of the method - for example, how were the molecular structures introduced in ADMET2.0? Did the Authors draw them separately, or prepared a list of SMILES?
I think the discussion would be easier to follow if the Authors incorporated some sort of a diagram, showing how they selected the potential antitbacterial compound 334 from the initial group of screened compounds, step by step.
Did the Authors consider other properties of studied compounds, e.g. skin permeability (this can be easily predicted, e.g. using SwissADME) - this can be important from the point of view of topical application.
Some minor improvements are possible, but generally speaking the manuscript is easy to follow.
Author Response
Thank you very much for your suggestion for my manuscript. I have already revised my manuscript according to your suggestion.
- The more detailed explanation of the method have added to the manuscript (line 332-335)
- The diagram gave added to the manuscript (line 137-146)
- The predicted skin permeability value have added to the manuscript (line 116 (Table 3); line 238 – 244). Please see the attachment

Reviewer 2 Report
This manuscript describes the use of virtual screening to identify from 378 Erythrina flavonoids those that have the best docking binding energy to GyrB ATPase subunit, lowest toxicity potential and best predicted pharmacokinetics property. Based on the virtual screening, the authors proposed that erystagallin B could be further developed as new antibacterial agent. The docking and in silico analysis appear to have been done according to proper procedures. In addition to reporting the binding affinities from the virtual screening, the authors should include images of some of the poses with the lowest binding energy and superimposed with the co-crystalized ligand so that the readers can see how the interactions compare with the co-crystalized ligand.
I do not have access to many of the references cited on Erythrina flavonoids. There are many publications that mentioned antibacterial activities of Erythrina flavonoids. The authors should report if the literature has mentioned any antibacterial activity or other biological activities for the Erythrina flavonoids reported as promising GyrB inhibitors based on the virtual screening conducted in this study.
Other Comments:
Abstract – “show potential as inhibitor of ATPase DNA gyrase B due to their lower binding affinity compared to inhibitor and ATP”; do authors mean lower binding affinity constant?
Line 308 – “hydrogen charges were added to the protein”; do authors mean hydrogen atoms and charges?
Supplementary Materials are mentioned at the end of the manuscript but not in the text. The supplementary table provided in response to this reviewer’s request should be mentioned in the text. The Table in Supplementary Materials should include the names and codes of all 378 flavonoids screened.
Minor edits of English language is recommended.
Author Response
Dear reviewer,
Thank you very much for your suggestion for our manuscript. We have revised our manuscript according to your suggestion
- The superimposed image of co-crystallized ligand and interaction with amino acids residue have added to the manuscript (line 164-179)
- The reported antibacterial activity of Erythrina flavonoids have added to the manuscript (line 86 – 94)
- Yes, the authors mean hydrogen atoms and charges (line 348)
- The supplementary table have mentioned in the manuscript (line 84-85)
